# Application of open domain adaptive models in image annotation and classification

Sheng Li[1], Zhousheng Chang[2]*, Haizhen Liu[3]

1 Innovation and Entrepreneurship Institute, Guangxi Normal University, Guilin, China, 2 Doctoral College, University for the Creative Arts, Epsom, United Kingdom, 3 Beijing Innovative Institute of Neodyna, Beijing, China

* czhousheng@163.com

## Abstract

In the field of computer vision, the task of image annotation and classification has attracted much attention due to its wide demand in applications such as medical image analysis, intelligent surveillance, and image retrieval. However, existing methods have significant limitations in dealing with unknown target domain data, which are manifested in the problems of reduced classification accuracy and insufficient generalization ability. To this end, the study proposes an adaptive image annotation classification model for open-set domains based on dynamic threshold control and subdomain alignment strategy to address the impact of the difference between the source and target domain distributions on the classification performance. The model combines the channel attention mechanism to dynamically extract important features, optimizes the cross-domain feature alignment effect using dynamic weight adjustment and subdomain alignment strategy, and balances the classification performance of known and unknown categories by dynamic threshold control. The experiments are conducted on ImageNet and COCO datasets, and the results show that the proposed model has a classification accuracy of up to 93.5% in the unknown target domain and 89.6% in the known target domain, which is better than the best results of existing methods. Meanwhile, the model check accuracy and recall rate reach up to 89.6% and 90.7%, respectively, and the classification time is only 1.2 seconds, which significantly improves the classification accuracy and efficiency. It is shown that the method can effectively improve the robustness and generalization ability of the image annotation and classification task in open-set scenarios, and provides a new idea for solving the domain adaptation problem in real scenarios.

## Introduction

In the digital age, the explosive growth of image data poses unprecedented challenges and opportunities for computer vision [1,2]. Image Annotation and Classification (IAC) has always been a key focus in the fields of industry and vision, and

**Data availability statement:** All relevant data are within the manuscript and its Supporting Information files.

**Funding:** The research is supported by National Social Science Foundation of China in 2022: Research on Evaluation System and Guarantee Mechanism of Labor Rights and Interests of Flexible Employees in Platform Enterprises (No.22XJY004). The funders had no role in study design, data collection and analysis, decision to publish, or preparation of the manuscript.

**Competing interests:** The authors have declared that no competing interests exist.

**Abbreviations:** DA, Domain Adaptation; ODA, Open Domain Adaptation; ODA-DTC, Dynamic Threshold Control - Open Domain Adaptation; P, Precision; R, Recall; AP, Average-Precision; DANN, Domain Adversarial Neural Networks; CADA, Conditional Adversarial Domain Adaptation; SEDA, Self-ensembling for Visual Domain Adaptation; L2T, Transfer Learning via Learning to Transfer; JAN, Joint Adaptation Networks; ADDA, Adversarial Discriminative Domain Adaptation.

is widely used in intelligent auxiliary processes such as intelligent driving, image retrieval, and healthcare. Its importance is self-evident [3]. The advancement of technology and the diversification of application scenarios have put forward higher requirements for the accuracy and generalization ability of IAC algorithms. The traditional IAC methods rely on manually designed features and shallow models, which have achieved some success in the early stages. However, due to the limitations of manual features and the simplicity of models, these traditional methods often perform poorly in complex practical application scenarios. Especially when facing large-scale datasets and inter-domain differences, its performance is limited by the ability to express features and generalization. With the growth of deep learning, image processing methods utilizing deep neural networks have begun to emerge, which can automatically learn complex feature representations and greatly perfect the performance of image processing tasks [4,5]. J. Li et al. designed a deep label specific feature learning model by combining deep convolutional networks and label embedding to enhance the alignment effect of specific labels in image classification [6]. This model could capture the dependency between image labels, improving the effectiveness of image classification. N. A. Koohbanani et al. proposed a Self-Path Model (SPM) combining self-supervised Convolutional Neural Networks (CNN) to obtain detailed information from high-resolution pathological images. When there was little or no available labeled data in the Target Domain (TD), SPM could improve the domain adaptability of tissue pathology image classification to achieve the goal of image depth detection [7]. S. Zhang et al. proposed a novel semantic fully supervised model by combining self-supervision and semantic features to improve the accuracy of existing 3D medical imaging tasks [8]. This model could effectively accelerate convergence and improve the accuracy of various 3D medical imaging tasks such as classification, segmentation, and detection. To further optimize the classification results of multi-modal image and text data, N. Xu et al. proposed a cross-modal image and text data classifier after combining sentiment analysis. The effectiveness of this classifier in improving the performance of multi-modal classification tasks was superior to traditional models [9]. Although the above research results can make technological improvements in their respective fields, even deep learning models may encounter insufficient generalization ability when applied to new and unprecedented domain data. In addition, most existing image processing methods focus on closed set conditions. In many real-world scenarios, there are significant differences between the distribution of training data and testing data, namely the Source Domain (SD) and TD, due to differences in collection conditions, heterogeneity of devices, or differences in data annotation methods. This leads to Domain Adaptation (DA) issues. Especially under open-set conditions, where the TD has new categories that have not been seen in the SD, traditional DA methods face significant challenges. Regarding this issue, domestic and foreign research has also explored it one after another. G. Chen et al. designed an open-set recognition accidental estimation model to improve the empirical classification performance of known data with open-set labeling. The proposed method was significantly superior to other existing methods and achieved advanced classification performance [10]. To improve the

recognition performance of pneumonia X-ray images in the open domain of medical images, Z. P. Jiang et al. constructed an improved VGG16 pneumonia image classification model. This model has produced superior test results compared to the current best practice CNN for medical image recognition [11]. A. Maracani et al. found that the open-set data collected by underwater plankton sensors in situ often has serious imbalances. Therefore, they designed a new transfer learning pipeline for plankton image classification by combining transfer learning. The average efficiency of open-set image classification transfer learning under this method was about 6% higher than other methods [12]. To improve the resolution effectiveness of ultrasound images of Papillary Thyroid Carcinoma (PTC), X. Ai et al. introduced an open dataset for capsule network training, and finally proposed an ultrasound image classification and diagnosis model [13]. The accuracy of PTC feature classification under this model was 81.06%, far exceeding other traditional methods. Parmar J et al. In order to enhance the effectiveness of open machine learning techniques in open set adaptive domain image classification, the research Confucians assembled natural language processing techniques and proposed a novel image classification method. The experimental results show that the accuracy of open set adaptive image classification under this method is higher than the traditional method and the image classification is more adaptive [14].Chen H et al. proposed a new method for stable classification of hyperspectral images, which is based on superpixel principal component analysis and random patch network. Not only can the data-driven approach be utilized, but it can also be applied to efficiently take into account more global and local spectral knowledge at the hyperpixel level. Test results on several open-set domain image classification datasets show that the method significantly outperforms several current state-of-the-art methods [15].

In summary, in recent years, image annotation and classification techniques play an important role in the field of computer vision, and are widely used in scenarios such as medical image analysis, intelligent surveillance and image retrieval. However, traditional supervised learning methods are highly dependent on the distribution of the training data, and when faced with the difference in data distribution between the source and target domains, especially in open-domain scenarios, the target domain data may contain new categories that have not been seen in the source domain, which results in insufficient generalization ability of the model and a significant decrease in classification accuracy. These limitations indicate that existing techniques are significantly inadequate for open-domain image annotation and classification tasks. In order to address the above issues, the scope of the research focuses on open-domain image annotation and classification tasks, specifically including the effective alignment of source and target domain features, the efficient utilization of unlabeled data in the target domain, the impact of dynamic threshold adjustment on classification performance, and the refinement of feature distributions in complex scenes. An optimized open-domain adaptive image annotation classification model (ODA-IAC) is proposed, whose main highlights include: the introduction of a domain adaptive algorithm (DA) to reduce the differences in feature distributions between the source and target domains; the combination of a dynamic threshold control module (DTC), which improves the model's classification accuracy of unknown categories by real-time adjusting of thresholds to adapt to the changes in the distribution of samples; and the adoption of a channel attention mechanism (CAM) to enhance the expression ability of important features; and adding the subdomain alignment strategy (SAS) to further refine the feature distribution within the target domain and improve the model's adaptability to complex scenes. The novelty of the research lies in the integrated use of dynamic threshold control, channel attention mechanism and subdomain alignment strategy, which solves the performance bottleneck of traditional domain adaptive algorithms in open-domain scenarios through the organic combination of modules. The expected contribution of the research provides an efficient method for processing images in unknown target domains, as well as the potential for wide application in practical applications.

## 1. Methods and materials

This study focuses on the domain differences in image classification, and proposes a dynamic threshold control DTC-ODA model by combining CAM and ODA algorithms. Secondly, for the problem of dubious classification results caused by the constant threshold value of the DTC-ODA model, the study continues to introduce the DWA and SAS methods for optimization, and finally proposes an optimized open-domain adaptive image annotation classification ODA-IAC model.

## 1.1 Construction of the ODA-IAC model

Traditional image annotation models typically use supervised learning to train on annotated datasets. However, due to domain differences between different datasets, these models often have poor generalization ability on new datasets [16]. Therefore, this study attempts to use existing annotation knowledge to label unknown data. DA algorithm is a type of machine learning algorithm aimed at solving the problem of mismatched data distribution between the SD and TD. The main reason for choosing the domain adaptive algorithm is that the significant difference in the data distribution between the source and target domains leads to the poor performance of traditional supervised learning methods in the target domain, especially in the open-domain scenarios, where the target domain contains new categories that have not been seen in the source domain, which makes the generalization ability of the model challenging. The domain adaptive algorithm solves the problem of inconsistent distribution of training and test data by aligning the feature distributions of the source and target domains, thus improving the classification accuracy of the model on the target domain [17-19]. Fig 1 is a schematic diagram of ODA.

In Fig 1, ODA can allow for the existence of unknown domains in the SD, and through appropriate strategies, it can solve the generalization problem of unknown TDs. However, general ODA may be limited by the quality and quantity of unlabeled TD data, which can affect the performance of the model. In open-domain scenarios, target-domain data usually contain unknown categories whose distribution is significantly different from that of the source domain. Fixed threshold models are difficult to balance the classification performance of known and unknown categories, which may lead to misclassification of unknown category data DTC effectively improves the model's ability to recognize unknown categories by dynamically adjusting the threshold according to the sample importance. Therefore, this study proposes a ODA-DTC model, whose structure is Fig 2.

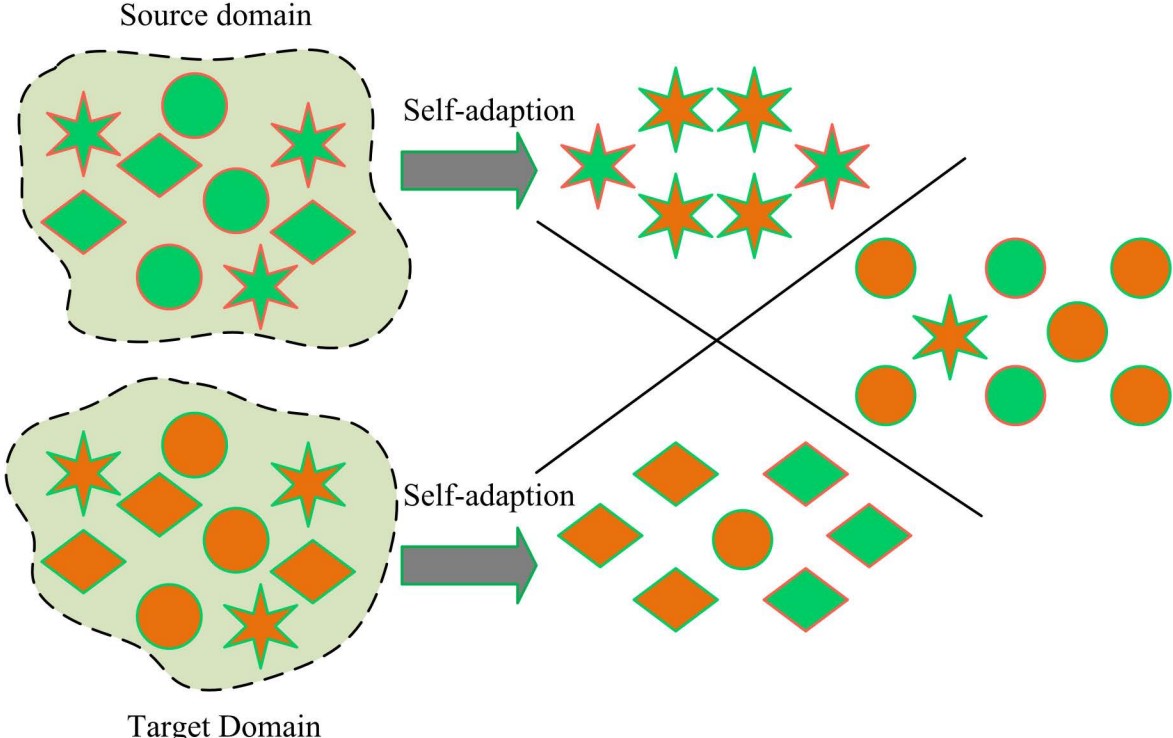

**Fig 1. ODA schematic diagram.**

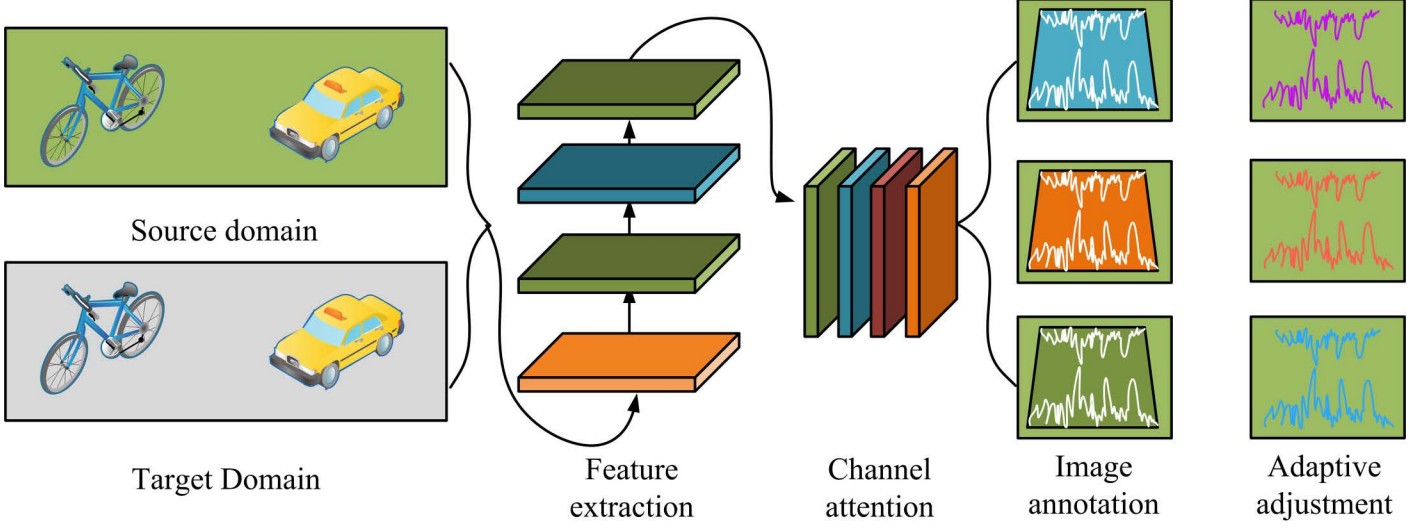

**Fig 2. ODA-DTC structural diagram.**

In Fig 2, DTC-ODA can be divided into four major modules, i.e., domain module, channel attention module, labeling module and adaptive module. The domain module is used to input source and target domain data for feature generation and distribution alignment; the channel attention module improves the expression of key features through weight addition; the annotation module realizes the accurate annotation of known and unknown categories through the dynamic thresholding mechanism; and the adaptive module dynamically adjusts the inter-domain differences in combination with the sub-domain alignment strategy to enhance the feature alignment effect of the target domain data. In terms of process, firstly, a convolutional neural network is utilized to extract multi-scale image features from low to high levels to provide rich basic information for the subsequent modules. Secondly, the channel attention mechanism is introduced to enhance the expression of important features through adaptive pooling and weighting operations to avoid the interference of redundant information on feature extraction. In addition, the dynamic threshold control module adjusts the feature extraction weights in real time according to the importance of the image samples, so as to optimize the expression ability of the features in diverse scenes. Fig 3 shows the CA structure.

In Fig 3, the whole module contains the original image layer (L1), pooling layer (L2), convolutional layer (L3), Sigmoid function layer (L4), channel weights layer (L5) and feature layer (L6). The original feature map is first input and processed through Adaptive Pooling (ATP), Average Pooling (AVP), and Maximum Pooling (MP), respectively. ATP uses 2D convolution for channel information learning. AVP and MP enhance the attention of local and global features. Then, the weights of each channel are calculated through 1D convolution and Sigmoid function, and finally, after weighting, the feature map is output [20]. It can be seen that CAM is able to weight different channels according to the importance of features, effectively enhancing the focus on key regions. Compared with the traditional fully connected layer or convolutional operation, this mechanism has stronger feature expression capability, especially superior in dealing with complex cross-domain data distribution. The calculation formula for channel weights is equation (1).

$$\omega = Sig \cdot Conv1D\left[(Avgpool(f)) + (Maxpool(f)) + (Adapool(f))\right] \tag{1}$$

In equation (1), *Sig* is the Sigmoid function. *Avgpool*, *Maxpool* and *Adapool* are AVP, MP, and ATP. *f* represents the original image, and its weighting calculation is equation (2).

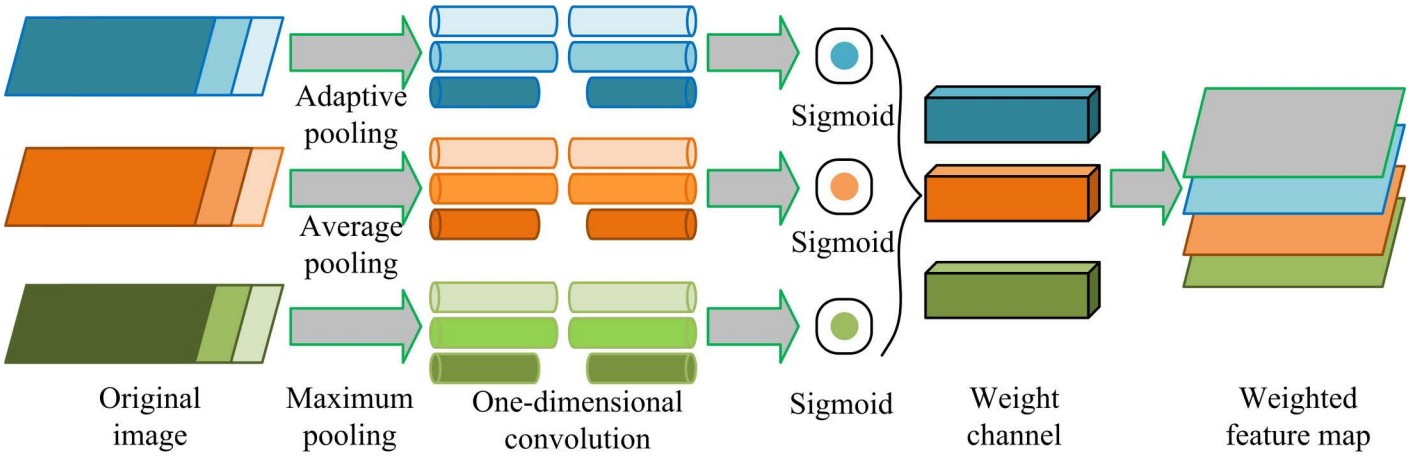

**Fig 3. Structure diagram of CA module.**

$$f' = \omega \times f \tag{2}$$

In equation (2), $f'$ represents the weighted original feature image. From the above calculation formula, compared to general attention modules, the CAM used in this study removes the fully connected layer and uses 1D convolution to effectively preserve the information exchange between channels while reducing the dimensionality of complex features. In addition, to improve the distribution similarity between the SD and the TD, this study first trains the model using labeled samples in the SD to learn image features in the SD. During this process, the loss function of the classifier is equation (3).

$$L_s(x_i^s, y_i^s) = \frac{1}{|D_s|} \sum_{x_i^s, y_i^s \in D_s} L_y\{C_y[G(x_i^s)], y_i^s\} \tag{3}$$

In equation (3), $x_i^s$ and $y_i^s$ respectively denote the $i$-th labeled image in the SD and the corresponding label for that image. $L_y$ and $C_y$ represent cross entropy loss and classifier, respectively. $D_s$ is the number of images in the SD. $G$ is a feature generator. After learning the feature information of label images in the SD, this study introduces a threshold $\beta$ to train the model. The threshold is determined by the $K$-th known label image and $K + 1$ unknown images in an adversarial game. If the discrimination probability of the unknown image is biased towards $\beta$, it indicates that the classifier has smaller distribution errors in the SD and TD, and the success rate of recognition will be higher. Specifically, in the open-domain scenario, the source and target domain data distributions may have large differences, especially the rare samples in the target domain are more likely to be neglected. Dynamic adjustment of weights By dynamically adjusting the training weights, the model is able to learn the rare sample features more efficiently, thus balancing the classification performance of known and unknown categories and improving the robustness of the model when dealing with unbalanced datasets. The calculation of error is equation (4).

$$L_{adv}(x_i^t) = -\beta \log\{p(x_i^t | y = K + 1)\} - (1 - \beta) \log\{1 - \beta(x_i^t) | y = K + 1)\} \tag{4}$$

In equation (4), $x_i^t$ represents the number of unlabeled images in the TD. $p(x_i^t | y = K + 1)$ is the probability that the $i$-th non-image in the TD belongs to the $K + 1$-th image category. To improve the alignment effect of image data between two domains, this study draws on the central idea of generative adversarial networks [21]. By generating adversarial samples,

the model is exposed to more diverse inputs during training, thus enhancing its generalization ability. This enhancement method not only improves the model's performance on known image classification, but also significantly improves its ability to handle unknown images. Through adversarial training, the generator generates realistic data samples, while the discriminator accurately distinguishes between real data and generated data as much as possible. This study assumes that the feature extraction network is a generator, with the SD being the real sample and the TD being the generated sample. By adversarial iteration between the final output sample of the generator and the discriminator, adaptive alignment between the two domains is ultimately achieved. At the same time, this study introduces a threshold $\psi$ as a constraint, defining that if the probability of the predicted sample image in the TD is greater than $\psi$, it is marked as a known label image. The constraint process is equation (5).

$$x_t = \begin{cases} x_{kt}, \psi(x_t \mid y = \hat{y}_i) \geq \psi \\ x_{unkt}, \psi(x_t \mid y = \hat{y}_i) < \psi \end{cases} \quad i = 1, 2, 3, \cdots, K$$

(5)

In equation (5), $x_t$ is the image in the TD. $\psi(x_t \mid y = \hat{y}_i)$ is the probability of the predicted sample image in the TD. $x_{kt}$ and $x_{unkt}$ are known and unknown images in the TD, respectively. In summary, this study proposes the operational process of the **DTC-ODA** model by combining the above types of modules and DTC, as shown in Fig 4.

Fig 4 illustrates the operational flow of the DTC-ODA model. The model contains 9 main links, i.e., source and target domain determination, feature generation, computation of weights, generation of feature maps, dynamic threshold judgment, feature updating, adversarial training, parameter updating, and image classification. Firstly, the data of source and target domains are fed into the feature generator for feature extraction, followed by the weighting operation of the generated feature map through the channel attention module to enhance the attention of important features. On this basis,

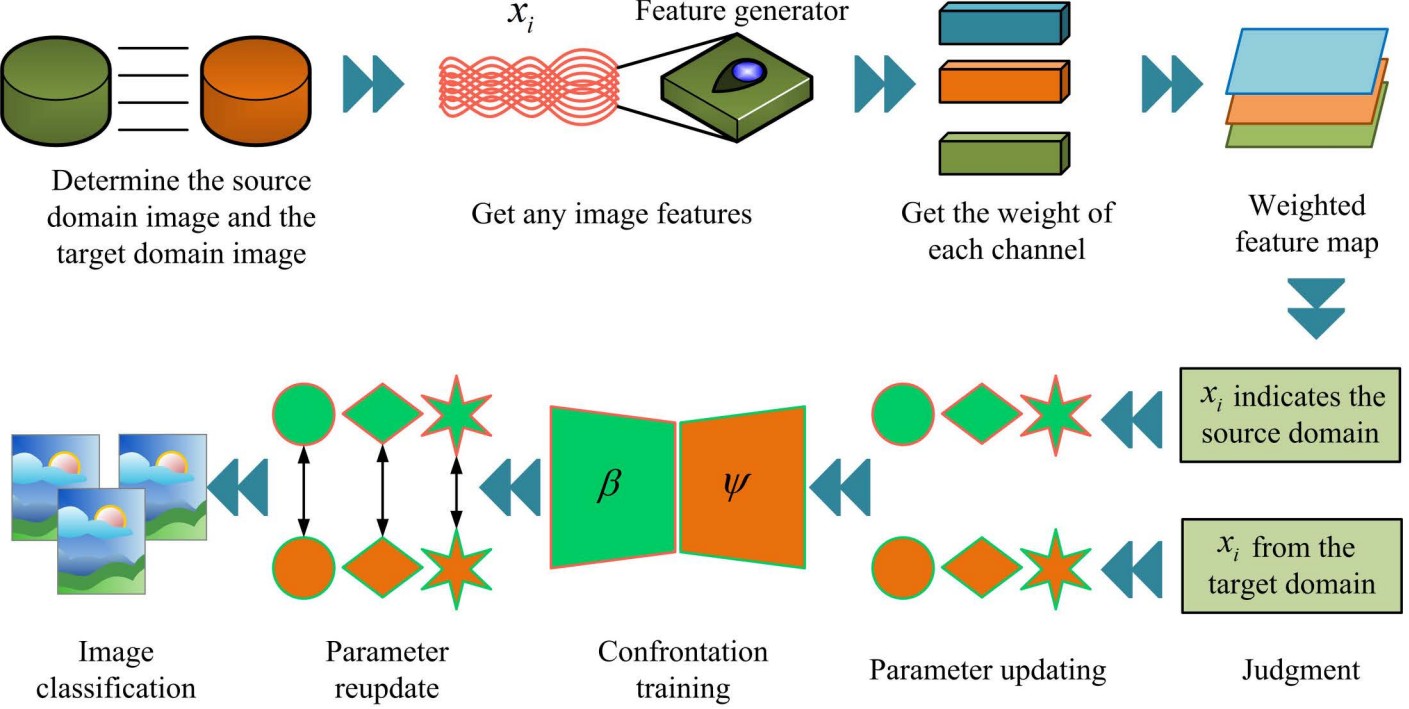

**Fig 4. The process of DTC-ODA model.**

the model decides the attributed category of the image according to its feature weights through the dynamic threshold judgment module, thus initially completing the classification of the target domain image. Subsequently, the model further optimizes the classification results through the feature update and adversarial training module to ensure the feature alignment effect between the source domain and the target domain, and performs image classification output at the end. In summary, by sharing the feature extraction layer, the multi-task learning strategy is able to fully utilize the complementary information between different tasks to further improve the overall performance of the model. Especially in unknown image classification, the multi-task learning strategy is able to utilize the boundary information of segmentation tasks to assist the classification decision. Finally, it makes the parameters that satisfy the threshold conditions used for image classification.

## 1.2 Optimization of the ODA-IAC model

The ODA-DTC image annotation model proposed in the previous section can effectively handle unknown images in the SD and TD and perform alignment processing. However, this threshold control method has certain limitations. For example, if the threshold cannot be flexibly adjusted according to the real-time training needs of the model, it may lead to a decrease in the reliability and validity of the training data results, and may cause negative transfer phenomena [22,23]. In view of this, this study introduces SAS, which divides open-set adaptive domains with a larger range and represents them in sub-domains. At the same time, weight parameters are introduced for DWA [24]. The optimized ODA-IAC model is shown in Fig 5.

In Fig 5, the optimization model can be segmented into 6 modules, namely source DM, target DM, feature extraction, Sub-domain Alignment (SDA), label classifier, and adaptive weights. SDA refers to segmenting a wide range of SD or TD and aligning features in a sub-domain manner to enhance attention to local feature information and improve the accuracy of image classification. The subdomain alignment module was chosen for the study because of its ability to capture feature differences between the source and target domains at a finer granularity, avoiding the problem of category confusion due to global feature averaging in full domain alignment. The schematic diagram of SDA is Fig 6.

Fig 6(a) and 6(b) are schematic diagrams of global alignment and SDA. By comparison, global alignment tends to focus more on the overall features of the image, ignoring the individual differences of different image samples in the domain, making the features of various types of images in the domain prone to confusion. SAS is more capable of displaying individual feature differences in images, mining obvious features for annotation and classification. The calculation formula for SAS is equation (6).

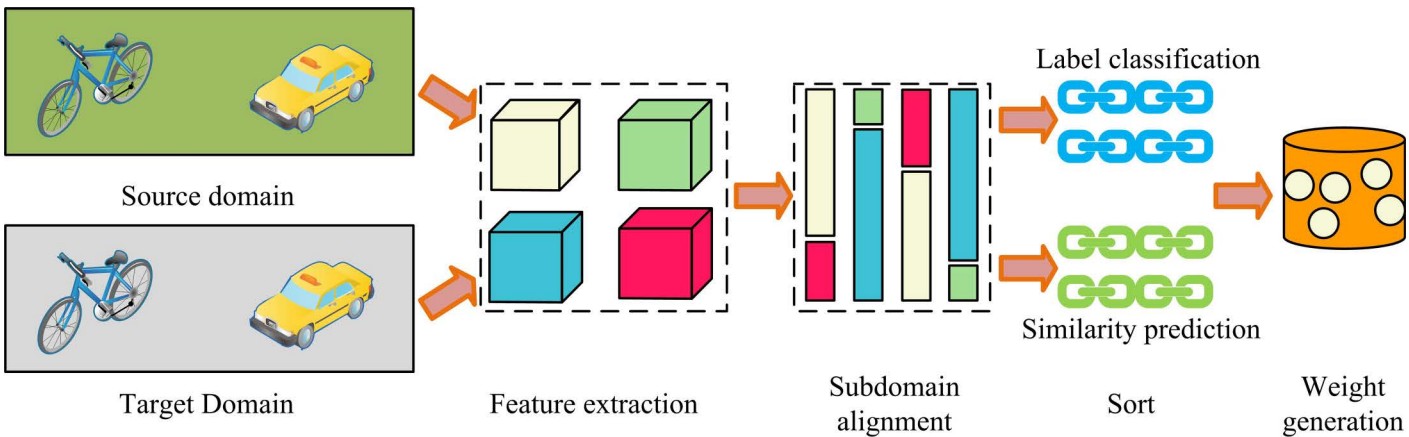

**Fig 5. ODA-IAC optimization model structure.**

$$L_{DA} = \frac{1}{n_s n_t} \sum_{i=1}^{ns} \sum_{j=1}^{nt} \| f(x_i^s) - f(x_j^s) \|^2 - \lambda \cdot D(f(x_i^s), f(x_j^s)) \tag{6}$$

In equation (6), $f(\_)$ represents the feature extraction function. $n_s$ and $n_t$ are the number of samples in the SD and TD. $D$ is difference measurement. $\lambda$ is the regularization parameter that balances these two objectives. $L_{DA}$ is the SDA loss function. Furthermore, considering that the core of SAS is to enhance the feature expression ability between internal individuals or sub-groups while maintaining overall domain alignment [25]. Therefore, this study optimizes the differences in sub-domain features, and the optimized results are shown in equation (7).

$$L_{sub} = \sum_{k=1}^{K} \| \frac{1}{|D_{S_k}|} \sum x_i \in D_{S_k} f(x_i) - \frac{1}{|D_{T_k}|} \sum x_j \in D_{T_k} f(x_j) \|^2 \tag{7}$$

In equation (7), $D_{S_k}$ and $D_{T_k}$ are the set of image samples in the $k$ -th sub-domain of the SD and TD. $L_{sub}$ is the optimization loss function for intra-domain differences. By optimizing the internal differences and optimizing the loss function, a more refined SAS can be achieved, which reduces the feature differences between different sub-domains while increasing the feature differences within the sub-domains to support more accurate image classification and annotation. In addition, the adaptive weight module is divided into two parts: similarity prediction and adaptive weight generation. Through this dynamic adjustment, the similarity reference value of known images and TDIs is improved, thereby avoiding negative transfer risks [26]. The loss function for similarity prediction is equation (8).

$$L_{s_1} = \frac{1}{n_s} \sum_{x_i, y_i \in D_s} CE \{ C_s [G(x_i)], y_i \} \tag{8}$$

In equation (8), $L_{s_1}$ is the loss function for similarity prediction. $x_i$ and $y_i$ are image samples and their image sample labels in the SD. $G(x_i)$ is the extracted features of image samples in the SD. $n_s$ is the total quantity of SDI samples. $CE$ means the cross entropy loss function. $C_s$ is the output result of similarity prediction. The similarity loss function generated by adaptive weights $L_{s_2}$ is equation (9).

$$L_{s_2} = \frac{1}{n_s} \sum_{i=1} \log \left\{ \sum_{c=1}^{K} C_S^c(G(x_i^s)) \right\} - \frac{1}{n_t} \sum_{j=1} \log \left\{ 1 - C_S^{K+1}(G(x_j^{ut})) \right\} \tag{9}$$

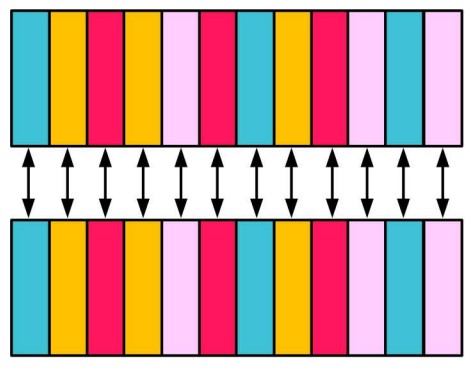

(a) Global alignment (b) Subdomain alignment

**Fig 6. Schematic diagram of sub-domain alignment.**

In equation (9), $\sum_{c=1}^{K} C_S^c(G(x_i^s))$ represents the known image probability of the top $K$ image samples in the SD in the similarity prediction stage. $C_S^{K+1}(G(x_j^{ut}))$ is the probability that an image in the TD is recognized as an unknown image of $K+1$. If $L_{s_1}$ and $L_{s_2}$ are closer to 1, it indicates that all image data in the SD will be successfully recognized. To improve the accuracy of image similarity comparison between the TD and the SD, this study attempts to pre-train the label classifier module and dynamically control its weight during the training process. The process is equation (10).

$$\begin{cases} S_1(x^t) = \sum_{c=1}^{K} C_S^c(G(x^t)) \\ S_2(x^t) = 1 - C_p^{K+1}(G(x^t)) \\ W(x^t) = (S_1(x^t)S_2(x^t)) \end{cases}$$

(10)

In equation (10), $S_1(x^t)$ and $S_2(x^t)$ are the discrimination probabilities of known images in the SD and known images in the TD in similarity prediction, respectively. $W(x^t)$ is the similarity weight during the training process. The size of this weight value can represent the similarity between the image sample in the TD and the known image. The larger the value, the closer the TDI is to the known image [27,28]. In addition, to meet the characteristics of large sample size and low accuracy in the early stage and small sample size and high accuracy in the later stage of model training, this study dynamically adjusts the threshold $W$, as shown in equation (11).

$$W = Sig\left[-\frac{1}{K}\sum_{i=1}^{K}\log(z_i^t)\right]$$

(11)

In equation (11), $z_i^t$ is the probability of a known image in the TD. By monitoring the training loss and gradient changes of the model in real time, the parameters of data enhancement are dynamically adjusted so that the enhanced samples are more in line with the learning needs of the current model. Based on the optimization of the above modules, this study proposes an ODA-IAC optimization model, as shown in Fig 7.

In Fig 7, the whole model has 11 segments. That is, determination of SD and TD, feature extraction, sub-domain alignment, calculating channel weights, SD similarity prediction, feature map generation, weight generation, threshold judgment, TD similarity calculation, parameter updating and image classification. First, the SDI data is given, and then the TDI data is input. After feature extraction by a feature extractor, it is separated into sub-domains. Subsequently, alignment operations are performed between sub-domains to transfer image features, making the distribution of domain features of the same type more similar. Finally, the label classifier module is used for label image recognition between the SD and TD, and the similarity between the TDI and the known image is compared using dynamic weights during the process.

## 2. Results

This study first used classification accuracy as an indicator to determine the optimal values of the three thresholds/weights in the model. At the same time, popular models of the same type were introduced for effectiveness comparison, and the Precision-Recall (PR) curves and areas of each model, as well as accuracy and recall, were tested. In addition, this study conducted visual comparison, confusion matrix comparison, and accuracy comparison between different models through simulation testing to verify the effectiveness and superiority of the research model.

### 2.1 Performance testing of optimized ODA-IAC models

In order to validate the model performance, the study uses ImageNet and COCO as the open-set dataset, and divides the data into training and test sets, where the training set accounts for 80% and the test set accounts for 20%. The training process was carried out in an Intel Core i7 CPU and NVIDIA GeForce GPU environment with 32 GB of RAM, developed

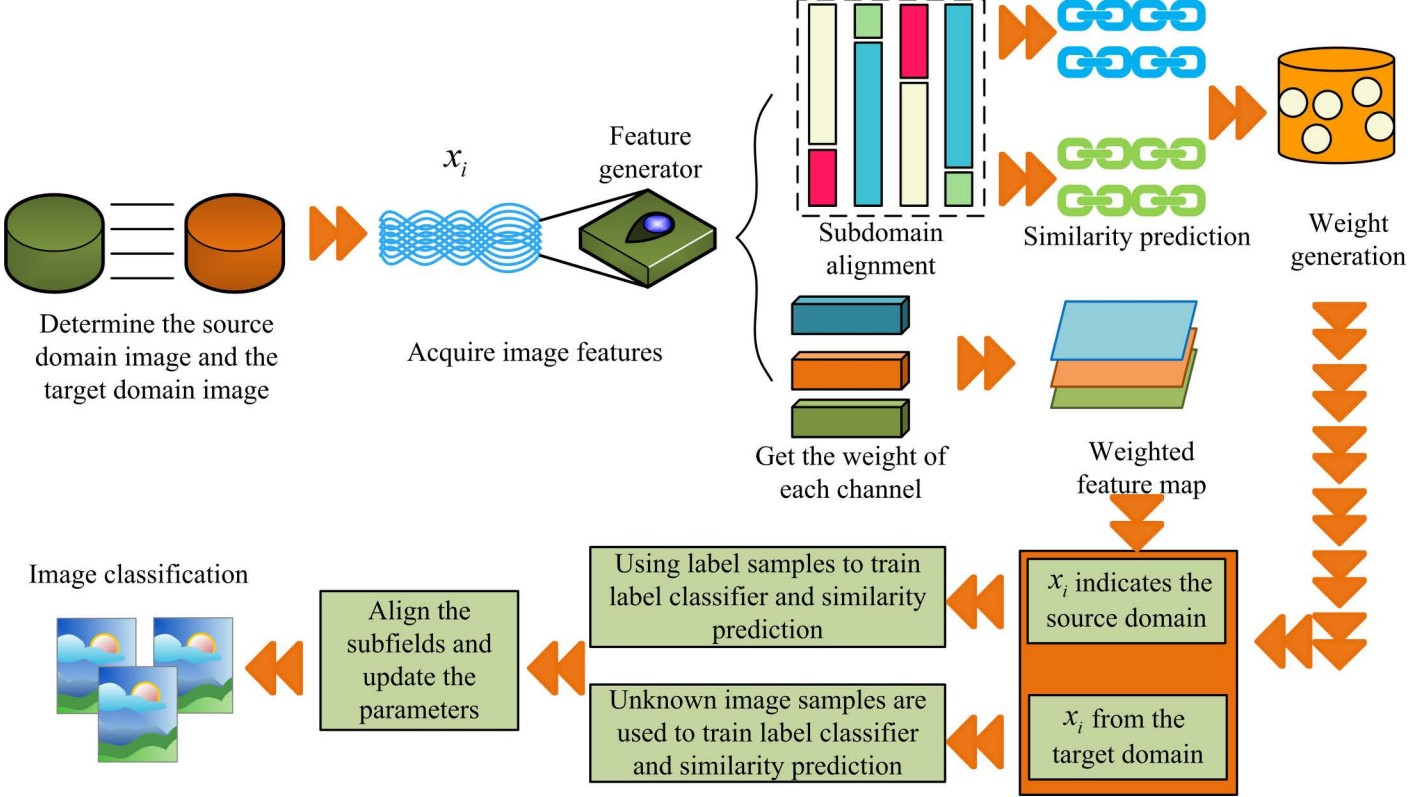

**Fig 7. Optimized ODA-IAC model flow.**

in Python, with a learning rate set to 0.002, a batch size of 64, and 200 training rounds. The model was trained using cross-entropy loss function and Adam optimizer through dynamic threshold adjustment and subdomain alignment optimization. In the testing phase, the model is evaluated with classification accuracy, checking accuracy, recall and F1 value, while comparing the performance of traditional domain adaptive models to ensure the comprehensiveness and reliability of the results. ImageNet and COCO datasets were used as open-set data sources. ImageNet is a large-scale image dataset containing over 14 million images, with hundreds to thousands of images per category. COCO contains over 330,000 images, each with instance annotations for multiple objects. It is a large-scale dataset used for tasks such as image recognition, object detection, and segmentation. In addition, the image specifications used in the study include RGB three-channel images with a resolution of 224×224 pixels in JPEG format, derived from ImageNet and COCO datasets. These images were normalized, including resizing and pixel normalization, to ensure adaptation to model input requirements. To determine the optimal values of thresholds $\beta$ and $\psi$ in threshold dynamic control, this study tested the Known Image Classification Accuracy (KICA) ACC-K and Unknown Image Classification Accuracy (UICA) ACC-UNK as test indicators. ACC-K measured the classification accuracy of the model on images of known categories. ACC-UNK measured the classification accuracy of the model on images of unknown categories. The results are shown in Fig 8.

Fig 8(a) shows the test results of threshold $\beta$, and Fig 8(b) shows the results of threshold $\psi$. In Fig 8(a), as $\beta$ increased, KICA gradually decreased while UICA continuously improved, indicating that the model's performance in IAC was gradually improving. When the two types of indicators intersected during this process, it indicated that the model performance tended to be balanced. At this time, the value of $\beta$ was 0.6, and ACC-K was 67%. In Fig 8(b), when $\psi$ was 0.7, the classification accuracy in the TD reached its peak first, with an ACC-UNK of 89%. When $\beta$ was 0.6, the known and

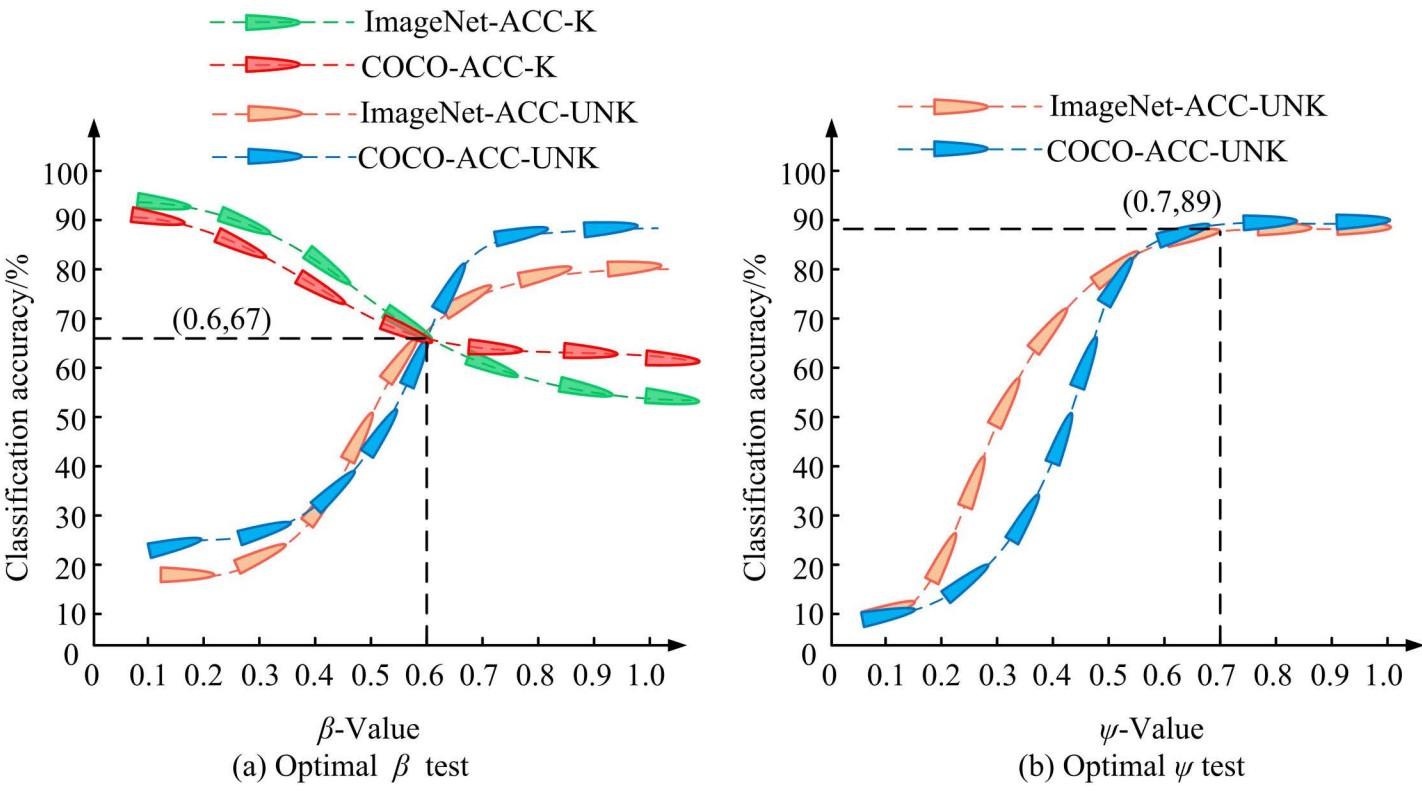

**Fig 8. Test results for optimal thresholds $\beta$ and $\psi$.**

unknown IAC performance of the model was more uniform. When $\psi$ was 0.7, the annotation classification performance of unknown images in the TD was the best. The study went on to test the dynamic weights $W$ under both types of data, which denote the weights that are dynamically adjusted according to the importance of the samples during the training of the model, as shown in Fig 9.

Fig 9(a) and 9(b) show the dynamic weight test results on the ImageNet and COCO datasets. In Fig 9, as the number of iterations of the model increased, the accuracy of model annotation classification gradually improved. In ImageNet and COCO, when iterated 180 and 250 times, the highest classification accuracy of the model was 83% and 88%. In the early stages of training, the number of known image samples for training was relatively small, so the larger the dynamic weight, the less impact the insufficient number of training samples would have. As the performance of the model gradually improved and stabilized, larger dynamic weights would actually suppress the training of the model. A moderate dynamic weight of 0.6 would balance training performance and sample consumption, making the model in the best performance state. This study continued to attempt to validate the performance of each module in the optimized model through ablation testing. Precision (P), Recall (R), and Average Precision (AP) were reference indicators. The AP metric measures the area of the model as a whole under the PR curve, i.e., the classification performance effect. The PR curve and its area with respect to the coordinate axis S were plotted, as shown in Fig 10.

Based on Fig 10(a)–10(d), the PR test curves and areas of the ODA algorithm, ODA-DTC, ODA-DTC-dynamic weight (ODA-DTC-DW), and ODA-DTC-DW-SDA algorithm were presented. The AP curve of the most basic ODA algorithm descended the fastest, with a PR of only 0.747. After adding channel attention, the performance of model image annotation classification was significantly improved, with a PR increase of 0.14. After continuing to introduce DWA and SAS, the model's recognition ability for unknown images was significantly improved by optimizing the model training and annotation

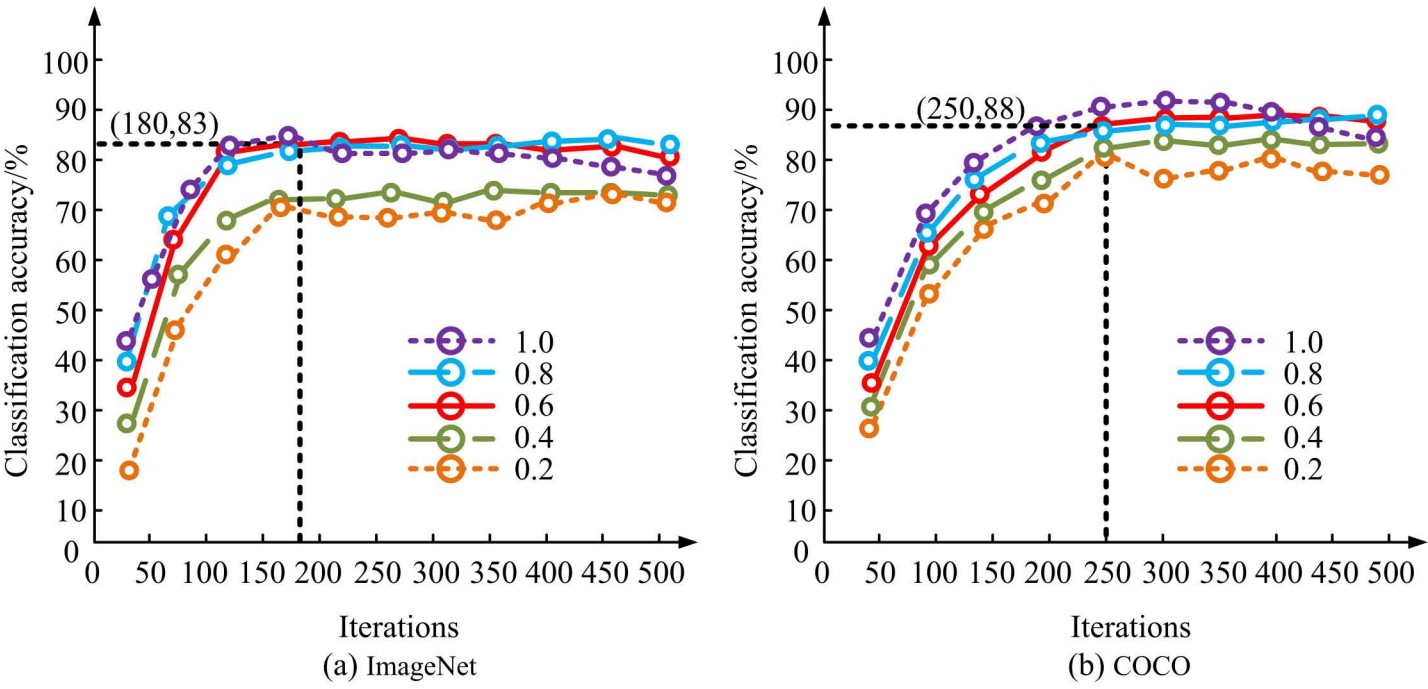

**Fig 9. Test results of dynamic weight $W$.**

classification process. The maximum PR at this time was 0.953. This study introduced the currently popular ODA-IAC models, such as Domain Adversarial Neural Networks (DANN), Conditional Adversarial Domain Adaptation (CADA), and Self-ensembling for Visual Domain Adaptation (SEDA) [29-31]. To evaluate the performance of the models in cross-domain data migration, the study conducts migration tests between the source and target domains, i.e., training the models on the source domain data and classifying the target domain images, as well as training the models on the target domain data and classifying the source domain images, respectively. Monte Carlo culling is also used to estimate the confidence of the different models to assess their reliability and generalization ability in cross-domain data migration. The confidence score reflects the confidence level of the model in dealing with unknown categories, which enhances the interpretability of the model. The test results are shown in Table 1.

In Table 1, the research proposed model performs well in the performance test of cross-domain data migration, and its check-accuracy, recall, and F1 value are significantly higher than those of the other models, both in the classification tasks of known and unknown domains. Among them, in the known domain, the study of the proposed model achieves 89.6% and 90.5% of check accuracy and recall, respectively, with an F1 value of 90%, an annotation consistency of 0.91, and the shortest running time of only 1.8 seconds. In the unknown domain, i.e., data that are unlabeled or not present at all in the training data of the source domain, the checking accuracy and recall are 89.3% and 90.7%, respectively, with an F1 value of 90.2%, and the annotation consistency is further improved to 0.92, with the shortest running time of only 1.2 seconds. In contrast, the performance of other models such as DANN and CADA is competitive, but they have longer running time and lower annotation consistency, especially in the migration test of unknown domains, with the highest F1 value of only 84.5%. In addition, the SEDA model performs relatively well in annotation consistency, but its F1 value and runtime still do not reach the level of the models mentioned in the study. Meanwhile, the confidence analysis further verifies the stability and reliability of the ODA-IAC model in dealing with unknown categories, and its confidence scores reach 0.91 and 0.93 in the known and unknown domains, respectively, which are significantly higher than those of the other models, indicating

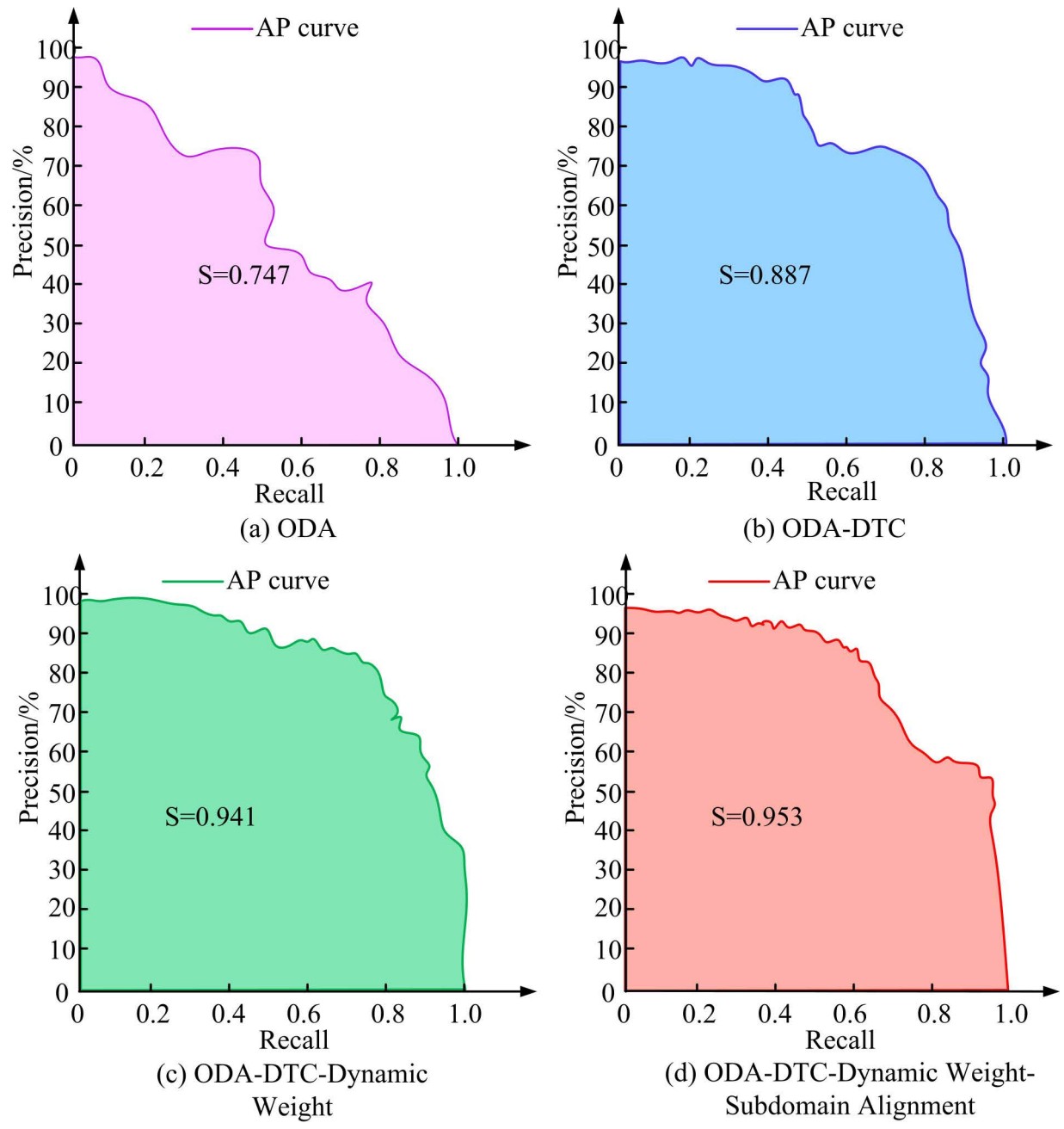

**Fig 10. PR curves and area test results for each module.**

that the model is not only able to provide high-precision categorization results, but also make predictions with higher confidence, which is crucial for reducing the risk of misclassification. In contrast, the confidence scores of DANN and CADA are 0.76 and 0.81, respectively, which are significantly less adaptable in the unknown domain, while the confidence score of SEDA reaches 0.87 but is still lower than that of ODA-IAC, suggesting that its handling of cross-domain data migration still has some limitations.

## 2.2 Simulation testing for the optimized ODA-IAC model

Similarly, to verify the simulation performance of the ODA-IAC, this study selected the VisDA dataset as the data source for simulation testing. The VisDA dataset contains both the source domain (synthetic images) and the target domain (real images), covering 12 widely distributed categories to maximize the diversity and representativeness between the source and target domains [32-34]. Fig 11 shows the example of VisDA's data sample.

The SDI was colorless and more modeled, while the TDI was more realistic and rounded. This study first introduced visual testing methods to compare image classification models similar to those used in ODA-DTC-DW-SDA, such as Transfer Learning via Learning to Transfer (L2T), Joint Adaptation Networks (JAN), and Adversarial Discriminative Domain Adaptation (ADDA) [35-37]. In order to gain a deeper understanding of how models classify in open-set scenarios, especially how they adapt to the transfer between source and target domains, the study introduces SHAP (SHapley Additive exPlanations) and LIME (Local Interpretable Model-agnostic Explanations) techniques to visualize the decision boundary for image classification, respectively. With SHAP and LIME, the study is able to accurately track how the model assigns different weights to different features when classifying images in the target domain, and is able to visualize the model's behavioral patterns when adapting to unknown categories. In particular, during the process of domain transfer, the model optimizes the classification performance by adjusting the decision boundary to ensure an appropriate response to the feature differences between the source and target domains. The test results are shown in Fig 12.

Fig 12(a)–12(d) show the visualization results of image type classification for L2T, JAN, ADDA, and research models. As can be seen in Fig 12, it is found that the classification decision boundary is looser when the methods of L2T and JAN are used, and the feature distributions of some target domain samples and source domain samples are not fully aligned, resulting in more misclassification cases of unknown categories. While the ADDA model has some improvement in domain alignment, its classification boundary still has a local offset, especially on the unseen category, and the feature mapping is more ambiguous. In contrast, the visualization results of the research-proposed ODA-IAC model on the VisDA dataset show that the domain alignment strategy significantly optimizes the classification boundaries, resulting in a tighter

**Table 1. Test results of indicators for different models.**

| Categorized object | Model | P/% | R/% | F1/% | Running time/s | Annotation consistency | Efficiency/% | Confidence score | Reference |
|---|---|---|---|---|---|---|---|---|---|
| Known areas | DANN | 78.6 | 81.2 | 79.9 | 4.2 | 0.86 | 89.72 | 0.79 | [29] |
| | CADA | 83.5 | 85.7 | 84.6 | 5 | 0.81 | 87.34 | 0.82 | [30] |
| | SEDA | 79.6 | 85.1 | 82.3 | 3.1 | 0.89 | 89.19 | 0.85 | [31] |
| | ODA-IAC | 89.6 | 90.5 | 90 | 1.8 | 0.91 | 91.28 | 0.91 | This study |
| Unknown areas | DANN | 82.1 | 84.6 | 83.3 | 4.2 | 0.85 | 79.34 | 0.76 | [29] |
| | CADA | 83.4 | 85.7 | 84.5 | 4.1 | 0.89 | 83.52 | 0.81 | [30] |
| | SEDA | 79.6 | 87.9 | 83.7 | 2.5 | 0.91 | 84.41 | 0.87 | [31] |
| | ODA-IAC | 89.3 | 90.7 | 90.2 | 1.2 | 0.92 | 89.56 | 0.93 | This study |

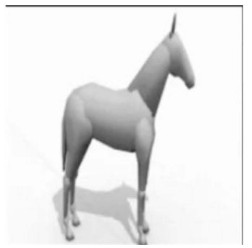 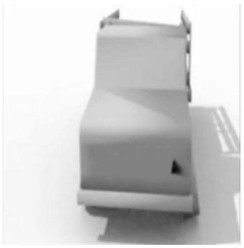 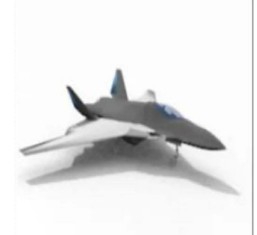 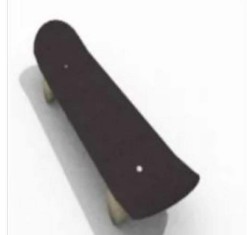

**Fig 11. Sample example of VisDA dataset.**

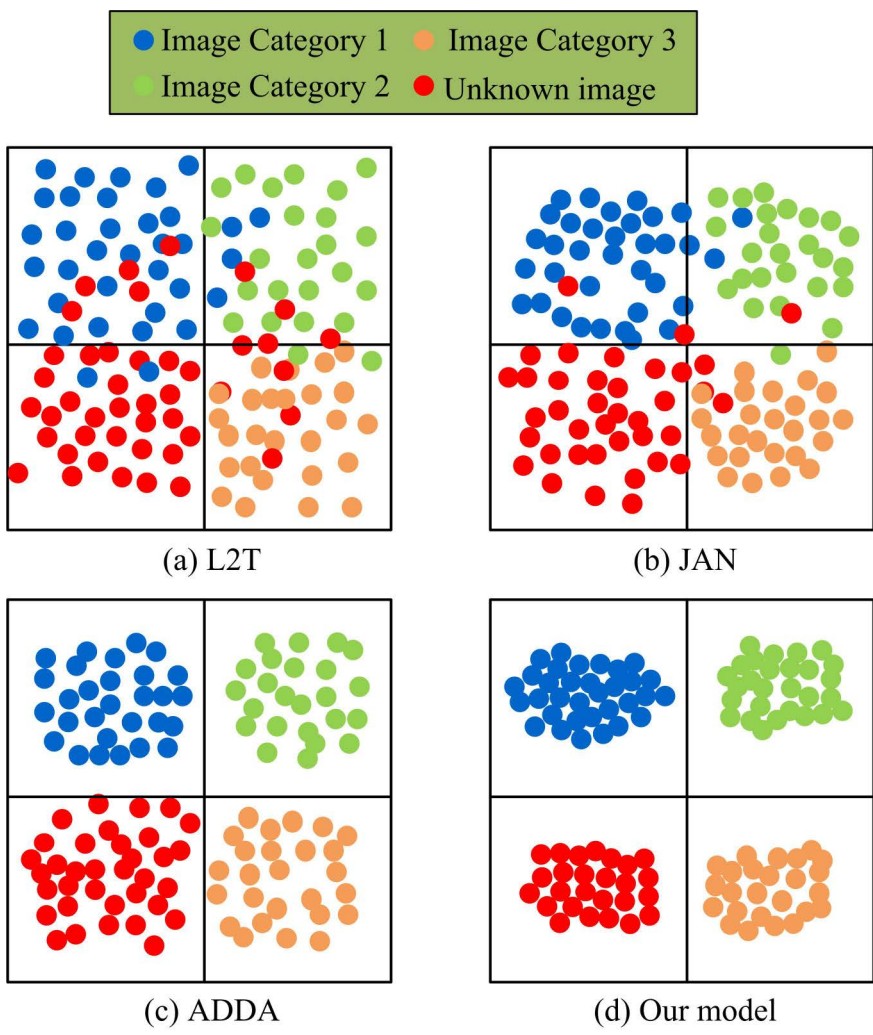

**Fig 12. Visual comparison results of different classification models.**

aggregation of features in similar categories and a more discrete distribution of features in different categories, which improves the model's adaptive ability on the target domain. The model prediction was compared with the actual results for a detailed evaluation of classification performance, as shown in Fig 13.

Fig 13(a) and 13(b) show the image confusion matrix results of ADDA and the research model. The ADDA model performed relatively average in image classification performance for 7 SD and TDs. There were 6 of them with a classification accuracy of 80 points, and only 4 with a score above 90 points. There were 7 research models with scores above 80 and 6 models with scores above 90. Although there were still some errors in the research model for recognizing and classifying a small number of different types of images, the overall performance was better than the ADDA model. The reason for this was that the research model adopted the DWA mechanism to adaptively adjust the alignment strategy between the SD and TD, thereby more effectively achieving feature alignment and classifier training between domains, and improving the model's generalization performance. The study used a larger real dataset as the source of the test data, namely Domain Adaptation Dataset (DomainNet).DomainNet contains images from six different domains, namely clip art, sketches, paintings, real photographs, information icons and sketches. In total, there are about 600,000 images,

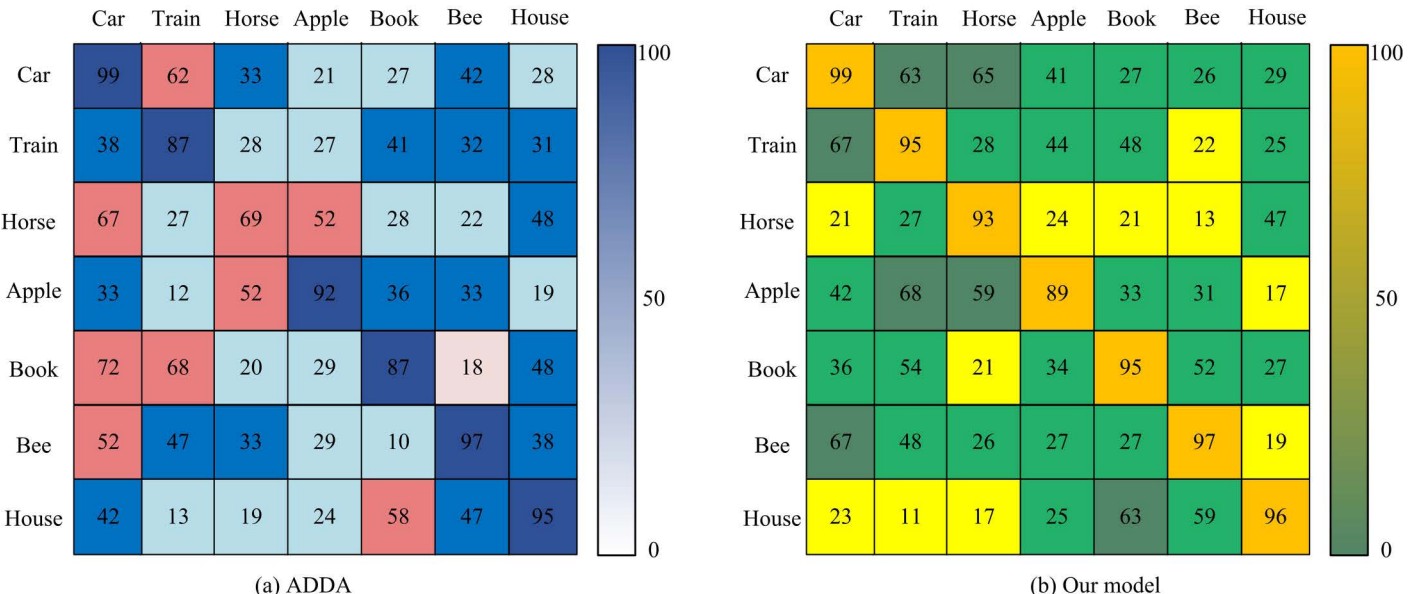

**Fig 13. Image confusion matrix results of two models.**

covering 345 categories, which include animals, daily objects, transportation, etc. The DomainNet is also introduced with a more advanced domain adversarial approach. The more advanced Domain-Adversarial Neural Network (DANN), Conditional Adversarial Domain Adaptation (CADA), and self-integrating for Visual Domain Adaptation (elf-) are also introduced. Ensembling for Visual Domain Adaptation (SEDA). ACC-K and ACC-UNK are used as indicators to test the migration effect of the four models between the source domain image and the target domain image under different lighting conditions, and the test results are shown in Table 2.

In Table 2, under low-light conditions, the classification accuracy of traditional methods such as DANN and CADA decreases due to feature instability, while the proposed model is optimized to optimize the feature alignment in the source and target domains through dynamic threshold adjustment, which enables the model to maintain a high level of classification performance in the target domain, and the ACC-UNK reaches 93.5%. Under normal lighting conditions, the ACC-K and ACC-UNK of the proposed model reach 93.2% and 92.8%, respectively, showing its strong adaptability in typical lighting environments, while under strong lighting conditions, the ACC-K and ACC-UNK still reach 90.9% and 91.5%, respectively, indicating that the dynamic threshold adjustment and sub-domain alignment strategies are able to effectively reduce the impact of the domain changes on the classification decision, making the source domain features more adaptable on the target domain, thus enhancing the generalization ability of the model in different environments. Furthermore, during the feature adaptation process between the source and target domains, the model aligns the data distributions of the source and target domains by introducing DA. Specifically, the features of the source domain are extracted and mapped to the feature space of the target domain through a feature generator and a channel attention mechanism. This process ensures that the features learned in the source domain can be effectively applied in the target domain by minimizing the feature differences between the source and target domains. To further optimize this adaptation process, the study employs SAS, which adapts the source domain features at a finer granularity by dividing the extensive source or target domains into multiple subdomains, allowing the features of the target domain to match the source domains more accurately and reducing the inter-domain distributional differences. Under strong light conditions, the ACC-K and ACC-UNK values of the proposed model reached 90.9% and 91.5%, respectively, which still dominated the performance compared with the other compared models, especially the ACC-UNK value of CADA decreased to 85.2% under strong light, while

**Table 2. Comparison of image transmission effects between domains under different lighting conditions.**

| Lighting conditions | Object | Model | ACC-K/% | ACC-UNK/% | Reference |
|---|---|---|---|---|---|
| Low light | Source Domain - Target Domain | DANN | 76.3 | 80.5 | [29] |
| | | CADA | 81.2 | 86.7 | [30] |
| | | SEDA | 86.7 | 90.1 | [31] |
| | | ODA-IAC | 89.6 | 93.5 | This study |
| | Target Domain - Source Domain | DANN | 74.1 | 81.2 | [29] |
| | | CADA | 79.6 | 85.7 | [30] |
| | | SEDA | 84.5 | 89.9 | [31] |
| | | ODA-IAC | 89.2 | 92.1 | This study |
| Normal light | Source Domain - Target Domain | DANN | 90.4 | 89.4 | [29] |
| | | CADA | 89.5 | 90.5 | [30] |
| | | SEDA | 91.6 | 91.3 | [31] |
| | | ODA-IAC | 93.2 | 92.8 | This study |
| | Target Domain - Source Domain | DANN | 89.8 | 91.2 | [29] |
| | | CADA | 91.1 | 92.4 | [30] |
| | | SEDA | 92.7 | 90.7 | [31] |
| | | ODA-IAC | 94.5 | 93.3 | This study |
| Strong light | Source Domain - Target Domain | DANN | 84.3 | 87.4 | [29] |
| | | CADA | 89.7 | 85.2 | [30] |
| | | SEDA | 90.2 | 89.3 | [31] |
| | | ODA-IAC | 90.9 | 90.2 | This study |
| | Target Domain - Source Domain | DANN | 89.4 | 88.2 | [29] |
| | | CADA | 88.3 | 89.3 | [30] |
| | | SEDA | 90.1 | 90.2 | [31] |
| | | ODA-IAC | 90.9 | 91.5 | This study |

the proposed model maintained a high stability under strong light conditions. Taken together, the proposed model can better adapt to changes in lighting conditions during cross-domain migration and shows stronger generalization ability and robustness, thanks to the effective combination of dynamic threshold adjustment and subdomain alignment strategy in the model, which results in finer feature extraction and better classification of images in different lighting environments. In addition, in order to adapt to real-time scenarios, the study integrates a parallel computing strategy, which ensures the stability and low latency of the model when processing high throughput data through GPU acceleration and distributed inference techniques.

## 3. Discussion and conclusion

### 3.1 Discussion

IAC is an important research direction in computer vision, and traditional IAC methods typically require numerous annotated data as training samples [38]. Given this, this study proposed a novel optimized ODA-IAC model by introducing an ODA model and combining CAM, DWA, and SAS. In the dynamic threshold adjustment test, a significant change relationship was observed between KICA and UICA by adjusting the threshold. Specifically, when the threshold was set to 0.6, KICA reached 67%, while UICA increased to 89%. This result indicated that by adjusting the threshold appropriately, the performance of the model on known and unknown image classification tasks could be effectively balanced, thereby improving the overall classification accuracy. This discovery aligned with the findings of Y Wei et al., which demonstrated the efficacy of threshold dynamic adjustment strategies in enhancing the efficacy of ODA models [39]. In addition, on the

ImageNet dataset, as the iterations increased, the model's classification accuracy could reach a peak of 88%. This test result highlighted the important role of DWA mechanism and SAS in optimizing the model training process, especially in terms of efficiency and accuracy when dealing with large-scale datasets. By comparison, the performance of the research model on ImageNet and COCO datasets was superior to other popular domain adaptive models, such as DANN, CADA, and SEDA. Especially when dealing with unknown TD data, research models exhibited higher accuracy and efficiency. This not only confirmed the effectiveness of the research model structure and strategy, but also formed consistency with the research results of advanced technology status, further proving the potential application of open-set domain adaptive models in IAC tasks. This result was consistent with W Jiao et al.'s research [40].

Although the research model has made significant progress in the accuracy and generalization ability of IAC, its performance in handling unknown TD data in extreme situations still needs to be optimized. For example, in cases of extremely imbalanced samples, the model performance may be affected. In addition, this study mainly focuses on static images, and its adaptability to IAC tasks in video or dynamic scenes has not been thoroughly explored. Future research directions can further explore and optimize the strategies of models in handling imbalanced samples and extreme unknown data to improve the model robustness. It is recommended that the model be applied to dynamic scenes or video data to ascertain its performance in such complex scenarios.

### 3.2 Conclusion

Aiming at the performance problems of traditional IAC methods, the study tried to improve the accuracy and generalization ability of IAC by developing a new ODA model. This study, based on ODA, introduced CAM, DWA, and SAS to construct a comprehensive framework for processing known and unknown image data, namely the optimized ODA-IAC model. The experimental data showed that when $\beta = 0.6$ and $\psi = 0.7$, the model had the best ACC-K and ACC-UNK values, which were 67% and 89%. When the dynamic weight $W = 0.6$, the model had the best classification accuracy in the ImageNet and COCO datasets, at 83% and 88%. Compared to the same type of model, it had a maximum P-value of 89.6%, a maximum R-value of 90.7%, a maximum F1-value of 90.2%, and a minimum time of 1.2s for labeling the classified target image. In addition, during simulation testing, the visual classification results of each image in the research model were the best, and the image classification was more compact. The highest ACC-K value for transferring SDI to TDI was 89.6%, and the highest ACC-UNK value for transferring TDI to SDI was 93.5%. The annotation classification results of this model for 7 types of images showed that there were 7 images with scores above 80 and 6 images with scores above 90. In summary, the proposed model significantly improves the classification performance on multiple open-set datasets. This indicates that the new model has significant advantages in terms of accuracy and generalization ability for the classification task of unknown TD data. However, this study also has certain limitations, such as in extremely complex domain conditions, the adaptability and accuracy of the model may still need further improvement. Future research will explore more strategies to optimize model structure and algorithms, especially performance optimization when dealing with extreme domain differences and large-scale datasets, to achieve wider applications and higher accuracy.

### Supporting information

**S1 File. Minimal data set definition**
(DOCX)

### Author contributions

**Investigation:** Haizhen Liu.

**Methodology:** Sheng Li, Zhousheng Chang.

**Resources:** Zhousheng Chang, Haizhen Liu.

**Validation:** Sheng Li.

**Writing – original draft:** Sheng Li, Haizhen Liu.

**Writing – review & editing:** Zhousheng Chang.

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
