## [Decision Letter · Decision Letter 0]

11 Dec 2024

PONE-D-24-38570Application of Open Domain Adaptive Models in Image Annotation and ClassificationPLOS ONE

Dear Dr. Chang,

Thank you for submitting your manuscript to PLOS ONE. After careful consideration, we feel that it has merit but does not fully meet PLOS ONE’s publication criteria as it currently stands. Therefore, we invite you to submit a revised version of the manuscript that addresses the points raised during the review process.

We look forward to receiving your revised manuscript.

Kind regards,

Shahul Hameed K A, PhD

Guest Editor

PLOS ONE

Journal Requirements:

Reviewers' comments:

Reviewer's Responses to Questions

**Comments to the Author**

1. Is the manuscript technically sound, and do the data support the conclusions?

Reviewer #1: Partly

Reviewer #2: Yes

2. Has the statistical analysis been performed appropriately and rigorously? 

Reviewer #1: N/A

Reviewer #2: Yes

3. Have the authors made all data underlying the findings in their manuscript fully available?

Reviewer #1: Yes

Reviewer #2: Yes

4. Is the manuscript presented in an intelligible fashion and written in standard English?

Reviewer #1: Yes

Reviewer #2: Yes

5. Review Comments to the Author

Reviewer #1: Abstract - Lack of Clear Motivation and Problem Definition, Unclear Novelty, Methodological Clarity, Ambiguity in Performance Metrics, Absence of Practical Implications and Language Precision and Flow.

Justification needed for "Authors might choose conventional methods over recent learning-based methods for several reasons, particularly when developing models for open-domain adaptive image annotation and classification. "

Some recent study can be included.

Highlights of works must be given.

Novelty should be highlighted.

Summary of limitations of the existing must be given.

Evaluate the model’s performance on data migration across domains, specifically when it is applied to classify target-domain images based on source-domain training and vice versa.

Assess consistency in annotations across similar images in both known and unknown domains to ensure stable performance.

Measure the time taken for the model to classify or annotate images, especially in unknown domains. Efficiency is particularly important if the model is intended for real-time applications.

In reviewer point of view "relying solely on classification accuracy may be insufficient because this metric alone does not provide a comprehensive view of the model's performance, especially in the context of open-domain and adaptive tasks. "

The model might perform well on certain types of domain shifts or specific datasets but may struggle with more complex, real-world shifts that introduce significant variability (e.g., varying lighting, resolutions, or object occlusions).

If the model is primarily tested on a few datasets with relatively mild domain shifts, its effectiveness in handling more extreme, real-world scenarios could be limited.

The model’s performance may be strongly tied to the quality and relevance of the source domain. If the source domain is significantly different from the target domain, or if it lacks diverse classes, the model might struggle with adaptation, resulting in poor generalization.

Data biases or imbalances in the source domain could impact how well the model performs when encountering less common or underrepresented classes in the target domain.

While the study may include comparisons with conventional methods, it might lack a thorough comparison with recent learning-based domain adaptation or open-set classification methods. This can make it difficult to assess how the proposed model truly stands up against state-of-the-art techniques.

The study may not address practical deployment issues, such as how well the model scales to large, real-world datasets or handles real-time annotation and classification needs.

Open-set recognition remains challenging, as the model must not only classify known images correctly but also accurately detect and separate unknown classes. False positives in unknown class detection can lead to reduced model reliability, while false negatives can compromise the model’s ability to recognize new patterns.

Reviewer #2: The manuscript has been drafted well.

The authors can consider the following comments

1. The specification of the Images used should be mentioned.

2. Feature extraction details have to be given elaborately.

3. Details about Training and testing the proposed model is missing.

4. Is the accuracy mentioned in table 2 an average accuracy?

5. The explanation about the proposed model can be even more descriptive

6. Clarify the scope of the research, including the specific aspects.

7. any specific Reason for 67% accuracy for Known images and 89% for unknown images. Usually Known images will have higher accuracy.

6. PLOS authors have the option to publish the peer review history of their article (what does this mean? ). If published, this will include your full peer review and any attached files.

**Do you want your identity to be public for this peer review?** For information about this choice, including consent withdrawal, please see our Privacy Policy .

Reviewer #1: No

Reviewer #2: No

---

## [Author Response · Author response to Decision Letter 1]

15 Jan 2025

Manuscript was revised according to comments.

---

## [Decision Letter · Decision Letter 1]

4 Feb 2025

PONE-D-24-38570R1Application of Open Domain Adaptive Models in Image Annotation and ClassificationPLOS ONE

Dear Dr. Chang,

Thank you for submitting your manuscript to PLOS ONE. After careful consideration, we feel that it has merit but does not fully meet PLOS ONE’s publication criteria as it currently stands. Therefore, we invite you to submit a revised version of the manuscript that addresses the points raised during the review process.

Be sure to:

Indicate which changes you require for acceptance versus which changes you recommendAddress any conflicts between the reviews so that it's clear which advice the authors should followProvide specific feedback from your evaluation of the manuscript

We look forward to receiving your revised manuscript.

Kind regards,

Shahul Hameed K A, PhD

Guest Editor

PLOS ONE

Journal Requirements:

Reviewers' comments:

Reviewer's Responses to Questions

**Comments to the Author**

1. If the authors have adequately addressed your comments raised in a previous round of review and you feel that this manuscript is now acceptable for publication, you may indicate that here to bypass the “Comments to the Author” section, enter your conflict of interest statement in the “Confidential to Editor” section, and submit your "Accept" recommendation.

Reviewer #1: All comments have been addressed

Reviewer #2: All comments have been addressed

2. Is the manuscript technically sound, and do the data support the conclusions?

Reviewer #1: Partly

Reviewer #2: Yes

3. Has the statistical analysis been performed appropriately and rigorously? 

Reviewer #1: Yes

Reviewer #2: Yes

4. Have the authors made all data underlying the findings in their manuscript fully available?

Reviewer #1: Yes

Reviewer #2: Yes

5. Is the manuscript presented in an intelligible fashion and written in standard English?

Reviewer #1: Yes

Reviewer #2: Yes

6. Review Comments to the Author

Reviewer #1: Please consider the following....

To make the model’s generalization more interpretable, uncertainty estimation methods like Monte Carlo Dropout or Bayesian Neural Networks can be used to show the model's confidence in predicting unknown classes. This helps in understanding how the model handles open-set situations.

Visualizing decision boundaries using techniques like SHAP or LIME can clarify how the model classifies images, particularly when adapting to domain shifts in open-set scenarios. The authors could also show how domain alignment affects the decision boundary or how features from the source domain are adapted for the target domain.

Reviewer #2: The authors responded for the quires. The review responses are satisfactory. The revised manuscript can be accepted for publication.

7. PLOS authors have the option to publish the peer review history of their article (what does this mean? ). If published, this will include your full peer review and any attached files.

**Do you want your identity to be public for this peer review?** For information about this choice, including consent withdrawal, please see our Privacy Policy .

Reviewer #1: No

Reviewer #2: **Yes: ** Nithya Rajagopalan

---

## [Author Response · Author response to Decision Letter 2]

11 Mar 2025

The manuscript has finished the revise.

---

## [Decision Letter · Decision Letter 2]

30 Mar 2025

Application of Open Domain Adaptive Models in Image Annotation and Classification

PONE-D-24-38570R2

Dear Dr. Chang,

We’re pleased to inform you that your manuscript has been judged scientifically suitable for publication and will be formally accepted for publication once it meets all outstanding technical requirements.

Kind regards,

Shahul Hameed K A, PhD

Guest Editor

PLOS ONE

Additional Editor Comments (optional):

Reviewers' comments:

Reviewer's Responses to Questions

**Comments to the Author**

1. If the authors have adequately addressed your comments raised in a previous round of review and you feel that this manuscript is now acceptable for publication, you may indicate that here to bypass the “Comments to the Author” section, enter your conflict of interest statement in the “Confidential to Editor” section, and submit your "Accept" recommendation.

Reviewer #1: All comments have been addressed

2. Is the manuscript technically sound, and do the data support the conclusions?

Reviewer #1: Yes

3. Has the statistical analysis been performed appropriately and rigorously? 

Reviewer #1: Yes

4. Have the authors made all data underlying the findings in their manuscript fully available?

Reviewer #1: Yes

5. Is the manuscript presented in an intelligible fashion and written in standard English?

Reviewer #1: Yes

6. Review Comments to the Author

Reviewer #1: Application of Open Domain Adaptive Models in Image Annotation and Classification" is well-structured and clearly conveys the focus of the study. It effectively highlights the use of adaptive models in a broad, open-domain setting, making it relevant to a wide audience interested in machine learning and computer vision.

7. PLOS authors have the option to publish the peer review history of their article (what does this mean? ). If published, this will include your full peer review and any attached files.

**Do you want your identity to be public for this peer review?** For information about this choice, including consent withdrawal, please see our Privacy Policy .

Reviewer #1: No

---

## [Editor Report · Acceptance letter]

PONE-D-24-38570R2

PLOS ONE

Dear Dr. Chang,

I'm pleased to inform you that your manuscript has been deemed suitable for publication in PLOS ONE. Congratulations! Your manuscript is now being handed over to our production team.

Kind regards,

on behalf of

Dr. Shahul Hameed K A

Guest Editor

PLOS ONE